# Iodine-catalyzed diazo activation to access radical reactivity

Pan Li[1,2], Jingjing Zhao[1], Lijun Shi[1], Jin Wang[2], Xiaodong Shi[2] & Fuwei Li[1]

Transition-metal-catalyzed diazo activation is a classical way to generate metal carbene, which are valuable intermediates in synthetic organic chemistry. An alternative iodine-catalyzed diazo activation is disclosed herein under either photo-initiated or thermal-initiated conditions, which represents an approach to enable carbene radical reactivity. This metal-free diazo activation strategy were successfully applied into olefin cyclopropanation and epoxidation, and applying this method to pyrrole synthesis under thermal-initiated conditions further demonstrates the unique reactivity using this method over typical metal-catalyzed conditions.

---

[1] State Key Laboratory for Oxo Synthesis and Selective Oxidation, Lanzhou Institute of Chemical Physics, Chinese Academy of Sciences, 730000 Lanzhou, Gansu, China. [2] Department of Chemistry, University of South Florida, Tampa, 33620 FL, USA. Correspondence and requests for materials should be addressed to X.S. (email: xmshi@usf.edu) or to F.L. (email: fuweili@licp.cas.cn)

Diazo compounds are versatile building blocks in chemical synthesis due to their ability to serve as carbene precursors under various metal-catalyzed conditions[1–6]. While typical metal carbenes are electrophilic, Zhang and other groups[7–9] reported a series of novel Co(II)-porphyrin carbene complexes with strong Co–C single-bond character, which displayed carbene carbon radical reactivity (Fig. 1a). Mechanistic investigation and new transformations have been reported[10–16], demonstrating the unique reactivity of these metal carbene radicals. Those seminal work disclosed the possibility of activating diazo compounds as radical precursors. However, as the general need in synthesis, new methodology is always highly desirable, especially those with significant different activation mode under distinct mechanism.

Herein, we report our discovery of iodine-catalyzed diazo activation and application of this method toward olefin cyclopropanation (Fig. 1b). Importantly, this method revealed a metal-free catalytic system to access diazo radical reactivity under mild conditions. Moreover, the ability to tolerate challenging substrates such as enamides ensure the successful synthesis of substituted pyrroles through cyclopropanation and sequential rearrangement in one pot, which highlights the advantage of this methodology over the typical metal-catalyzed system.

## Results

**Combining iodine-catalyzed diazo activation and photocatalysis.** This iodine-catalyzed diazo activation originated from an accidental discovery[17–20]. It has been recently reported that aryl diazonium salts ($ArN_2^+$) are effective oxidants in promoting gold(I) oxidation to gold(III)[21–23]. To extend the reactivity to alkyl diazonium salt, we hypothesized that treating diazo compound with proper electrophile could generate a reactive diazonium intermediate in situ, which might serve as a potential oxidant for gold(I) oxidation. To explore the feasibility of this idea, we investigated reactions of ethyl diazoacetate (EDA) **1a** with various electrophiles. Interestingly, treating EDA **1a** with $I_2$ led to the formation of ethyl diiodoacetate **1a'** in nearly quantitative yield even at room temperature, suggesting rapid decomposition of transient intermediate **A** (Fig. 2a). Notably, Suero and coworkers[24] recently reported an interesting styrene cyclopropanation with $CH_2I_2$ under photoredox conditions. As proposed by the author, the iodo-substituted methyl radical was formed via a photoredox reductive quenching cycle, which subsequently reacted with an alkene to form radical intermediate **B**. However, access reductants ($i$-Pr$_2$EtN and Na$_2$S$_2$O$_3$) are required to initiate the reductive Ru$^I$/Ru$^{II}$ cycle and scavenge the resulting $I_2$ (Fig. 2b).

With the realization of diiodo compound **1a'** by reacting diazo compound with $I_2$ under mild conditions, we wondered whether this process could be combined with photo activation to access diazo radical reactivity. Notably, it has been reported that electron-deficient alkyl halides could serve as effective oxidants to convert $[Ru(bpy)_3]^{2+}$ into $[Ru(bpy)_3]^{3+}$ under photochemical conditions[25–29]. Thus, it is reasonable to expect **1a'** to serve as an oxidant and promote oxidative Ru$^{II}$/Ru$^{III}$ cycle (Fig. 2c). Addition of the resulting carbon radical **C** to alkene will form the radical intermediate **B'**, which undergoes cyclization through the release of an iodine radical. The iodine radical would react with iodide to form anionic radical $I_2^{-\bullet}$. Ultimately, $[Ru(bpy)_3]^{3+}$ acquires an electron from $I_2^{-\bullet}$ to regenerate $I_2$ and $[Ru(bpy)_3]^{2+}$. Compared with the process reported by Suero and coworkers[24], this proposed approach is more atom economic since only catalytic amount of iodine could initiate the radical process to avoid the usage of access reductants ($i$-Pr$_2$EtN and Na$_2$S$_2$O$_3$).

**Olefin cyclopropanation under photo-conditions.** To our delight, the desired cyclopropanation product **3a** was obtained in 96% yield (dr = 1.9:1, diastereomeric ratio) using the combination of $I_2$ (10 mol%) and Ru(bpy)$_3$Cl$_2$ (1 mol%) as catalysts (Table 1). Control experiments clearly demonstrated that iodine and photo catalysts were indispensable for this transformation. The reaction worked well with either blue or white light, while no product **3a** was obtained if reacting in the dark, which is consistent with the proposed photo-initiated mechanism. Ir(dtbbpy)(bpy)$_2$PF$_6$ could also catalyze this reaction with 65% yield, while organo-photosensitizer eosin Y could not promote this transformation.

Diiodide compound **1a'** could substitute $I_2$ as an effective catalyst for this transformation, giving **3a** in 94% yield. This result clearly supported our hypothesis that **1a'** is the key intermediate for photo-initiation. No **3a** was observed when using $Br_2$ instead of $I_2$ as catalyst[30–32]. Interestingly, reaction of alkene and **1a'** under photo activation conditions (no diazo **1a** and $I_2$) gave only trace amount of product. This is likely caused by the formation of large amount of $I_2$ (without the addition of extra reductants such as Na$_2$S$_2$O$_3$), which would quench the photo-excited state of the catalyst *[Ru(bpy)$_3$]$^{2+}$ and prevent photocatalytic cycle from happening[33]. Conducting the reaction (**2a** and **1a'**) under Suero's conditions (with $i$-Pr$_2$EtN and Na$_2$S$_2$O$_3$ in MeCN) gave the desired product **3a** in 35% yield[24]. These results highlighted the overall high efficiency of this photo-catalyzed Ru$^{II}$/Ru$^{III}$ oxidative cycle by simply using diazoacetate and catalytic iodine. (2,2,6,6-Tetramethylpiperidin-1-yl)oxyl or (2,2,6,6-tetramethylpiperidin-1-yl)oxidanyl (TEMPO) could effectively quench the reaction, which was consistent with the proposed radical mechanism.

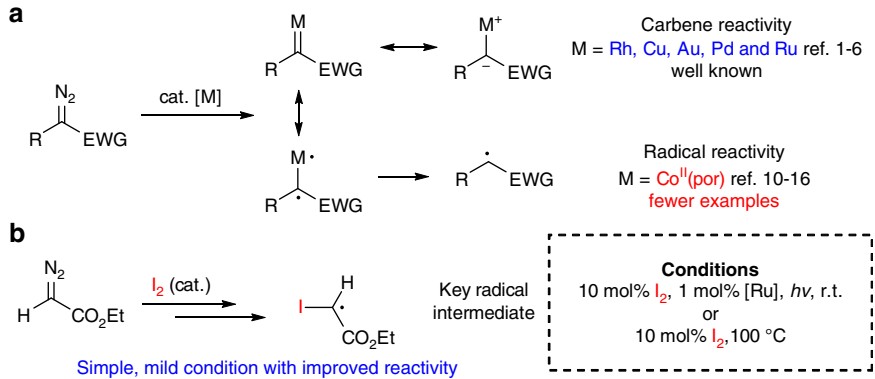

**Fig. 1** Catalytic diazo compound activation: carbene vs. radical. **a** Metal-catalyzed diazo compound activation. **b** This work: access diazo radical reactivity through simple iodine catalysis

Interestingly, when conducting the reaction with olefin **2b**, <10% **2b** conversion was observed with 10 mol% $I_2$ as catalyst. Analyzing the reaction mixture revealed the formation of diiodide compound **4** as the major product through radical-promoted cyclopropane ring opening. Increasing the amount of $I_2$ to 1 equiv. gave the desired diiodide product **4** in 75% isolated yield (E/Z = 15:1) (Fig. 3). These results not only demonstrated formation of radical intermediates **D** and **E** in a typical radical clock experiment, but also revealed faster iodination of radical **E** over either hydrogen radical elimination or radical cyclization[34].

With the optimized conditions revealed, we investigated the reaction scope under photo-initiated conditions. The results are summarized in Table 2. Styrenes bearing both electron-donating and electron-withdrawing groups are all suitable for this transformation, giving desired products (**3a**–**3h**) in good to

excellent yields (around 2:1 dr ratio). Other arene-substituted olefins also works well (**3i**, **3k**–**3m**). However, the reactions only gave diminished yields for highly electron-deficient alkenes, such as 2-vinylpyridine, chalcone, and p-NO$_2$ styrene. In addition, messy reactions were observed for diene, methyl 2-phenylacrylate, and vinyl ether substrates.

**Iodine-catalyzed diazo activation toward olefin cyclopropanation under thermal conditions**. Although this iodine-catalyzed diazo activation under photo-initiation provided a simple and novel approach to access diazo radical reactivity, it only displayed a limited substrate scope. Control experiments disclosed two main reasons: (1) Ru(bpy)$_3$Cl$_2$ would be quenched by certain alkenes, such as vinylferrocene, p-NO$_2$-styrene and enamide; (2) some alkenes (vinyl ether and dienes) would decompose under photo-conditions (see details in Supplementary Figs. 2 and 3). To further expand this methodology to more challenging substrates such as photo-sensitive alkenes and electron-deficient alkenes, we wondered whether alternative ways to generate radical intermediate **C** is feasible. We hypothesized this could be achieved from C–I bond homo-dissociation of diiodide compound **1a′** at elevated temperatures. Unfortunately, heating reaction mixtures of **2a** and **1a′** at 100 °C gave no conversion of alkene **2a** after 24 h (Table 3, entry 1), suggesting no formation of radical **C**[35]. The conversion of **2a** was <10% while reacting with diazo **1a** under simple heating conditions[36]. To our delight, adding 10 mol% $I_2$ into reaction mixtures of **2a** and **1a** at 100 °C successfully delivered the desired cyclopropane **3a** in nearly quantitative yield (entry 3). Interestingly, increasing the amount of $I_2$ to 1 equiv. completely quenched the reaction (entry 4), suggesting the rapid formation of **1a′** when treating **1a** with $I_2$, which is not a valid substrate alone for this reaction as shown in entry 1. Furthermore, switching $I_2$ catalyst to 10 mol% **1a′** gave same excellent yield of **3a** (entry 5). These results clearly demonstrated that reaction between diiodide compound **1a′** and diazo **1a** at elevated temperature produced the active intermediate that promoted the overall transformation. In addition, only trace amount of product

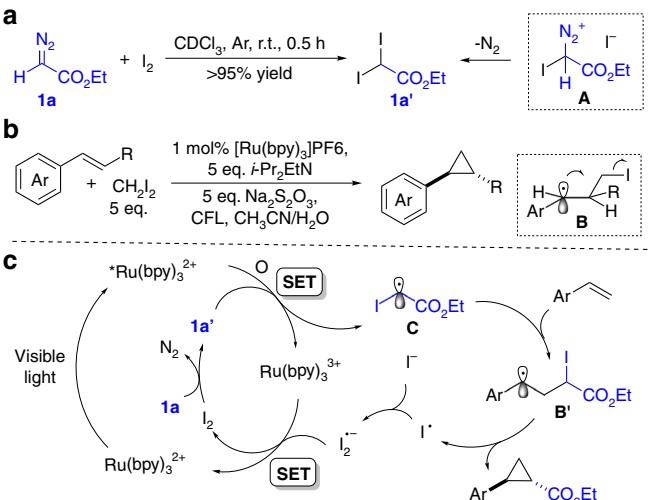

**Fig. 2** Synthesis of diiodoacetate **1a′** and our hypothesis. **a** Rapid reaction between diazo **1a** and $I_2$. **b** Olefin cyclopropanation via photo-initiated CH$_2$I$_2$ activation reported by Suero. **c** Our proposal: combining iodine promoted diazo activation and photo-initiation

**Table 1 Optimization of the reaction conditions under photo-initiated conditions**

| Alternation from above conditions | 2a conv. | 3a yield |
|---|---|---|
| None | 100% | 96% (92%) |
| No $I_2$ | 0% | ND |
| No Ru(bpy)$_3$Cl$_2$ | 0% | ND |
| White light instead of blue light | 95% | 85% |
| In the dark | 0% | ND |
| Ir(dtbbpy)(bpy)$_2$PF$_6$ instead of Ru(bpy)$_3$Cl$_2$ | 75% | 65% |
| Eosin Y instead of Ru(bpy)$_3$Cl$_2$ | 0% | ND |
| 10 mol% **1a′** instead of $I_2$ | 100% | 94% |
| 10 mol% Br$_2$ instead of $I_2$ | 0% | ND |
| No $I_2$, 1 equiv. of **1a′** instead of **1a** | <5% | Trace |
| No $I_2$, 1 equiv. of **1a′** instead of **1a** i-Pr$_2$EtN (1 equiv.), Na$_2$S$_2$O$_3$ (1 equiv.),MeCN instead of DCE (Suero's condition, Ref. [22]) | 55% | 35% |
| Addition of 1 equiv. of TEMPO | 0% | ND |

Reaction conditions: 1mol% photocatalyst and 10 mol% $I_2$ were added to a DCE (3 mL) solution of **2a** (0.3 mmol) and EDA **1a** (0.3 mmol), and reaction was kept under argon (degassed) at room temperature under blue light for 24 h. Conversion and yield were determined by $^1$H NMR spectroscopy using 1,3,5-trimethoxybenzene as internal standard. Isolated yield are given within parentheses. *EDA* ethyl diazoacetate, *DCE* 1,2-dichloroethane, *r.t.* room temperature, $^1$H NMR proton nuclear magnetic resonance, *TEMPO* (2,2,6,6-tetramethylpiperidin-1-yl)oxyl or (2,2,6,6-tetramethylpiperidin-1-yl) oxidanyl, *ND* not determined

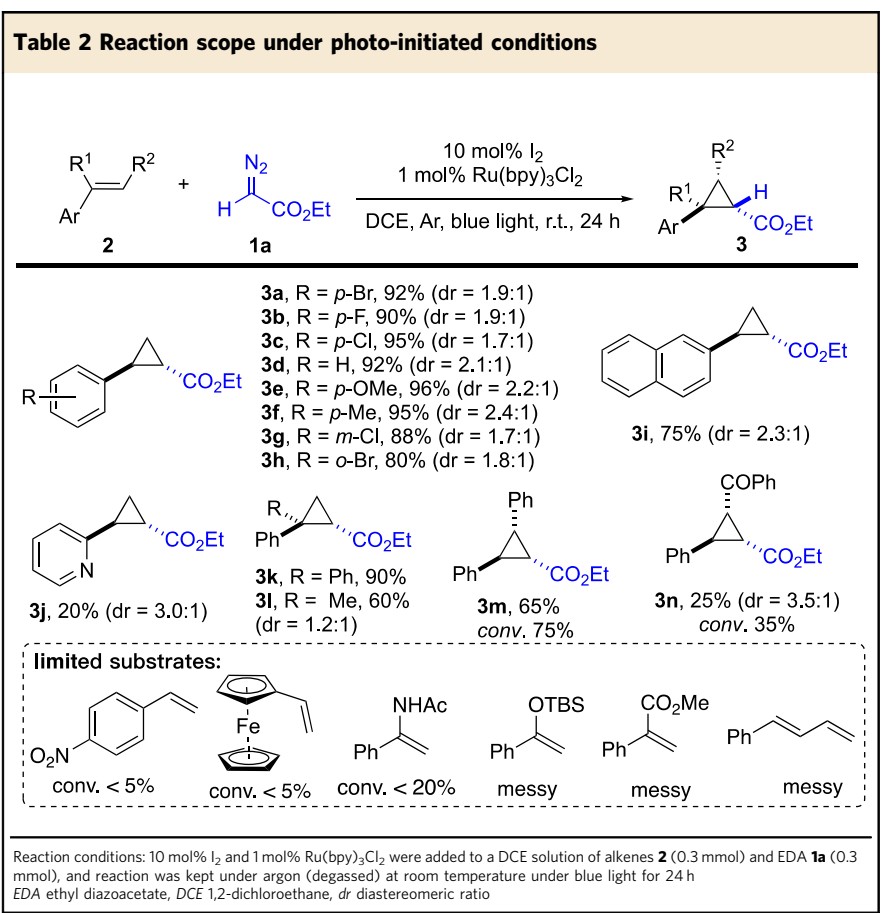

**Fig. 3** Radical clock experiment. Proposed mechanism for the synthesis of **4**

**Table 2 Reaction scope under photo-initiated conditions**

Reaction conditions: 10 mol% $I_2$ and 1 mol% Ru(bpy)$_3$Cl$_2$ were added to a DCE solution of alkenes **2** (0.3 mmol) and EDA **1a** (0.3 mmol), and reaction was kept under argon (degassed) at room temperature under blue light for 24 h
*EDA* ethyl diazoacetate, *DCE* 1,2-dichloroethane, *dr* diastereomeric ratio

**3a** was formed at 80 °C, and other catalysts (Br$_2$, KI, NIS) were invalid for this transformation.

To explore the reaction mechanism under the thermal conditions, several control experiments were performed as shown in Fig. 4. First, similar to photo-initiated conditions, reaction was completely quenched by TEMPO and intermediate **G** was successfully detected by gas chromatography–mass spectrometry (Supplementary Fig. 1). Radical clock experiment give the desired diiodide product **4** with 80% yield ($E:Z = 4:1$). Importantly, electron paramagnetic resonance (EPR) experiments were performed, and, fortunately, both nitrogen radical and carbon radical were clearly detected (see EPR spectra in Supplementary Fig. 4). Although the exact mechanism requires further investigation, based on current experimental results, it is reasonable to speculate the formation of active reaction intermediate

upon mixing **1a** and **1a′** (such as **H**). This intermediate would generate nitrogen radical **I** and/or carbon radical **C** via single electron transfer (SET) process at elevated temperature. Ultimately, the cyclopropane was formed through intramolecular radical substitution as shown in Fig. 2.

Encouraged by the simple and metal-free diazo activation method under thermal conditions, we explored the reaction scope, especially the challenging substrates that photo-initiated method failed to work. The results are summarized in Table 4. Compared with photo-initiated conditions, the thermal-initiation reactions were significantly cleaner, giving higher yields in almost all cases (i.e., **3j** from 20 to 94%, **3l** from 60 to 92%). Moreover, alkene substrates that are not suitable for photo-conditions, such as 1-nitro-4-vinylbenzene and vinylferrocene, worked well under the thermal conditions, giving the desired products **3p** and **3t** in

**Table 3 Optimization of the reaction conditions under thermal-initiation conditions**

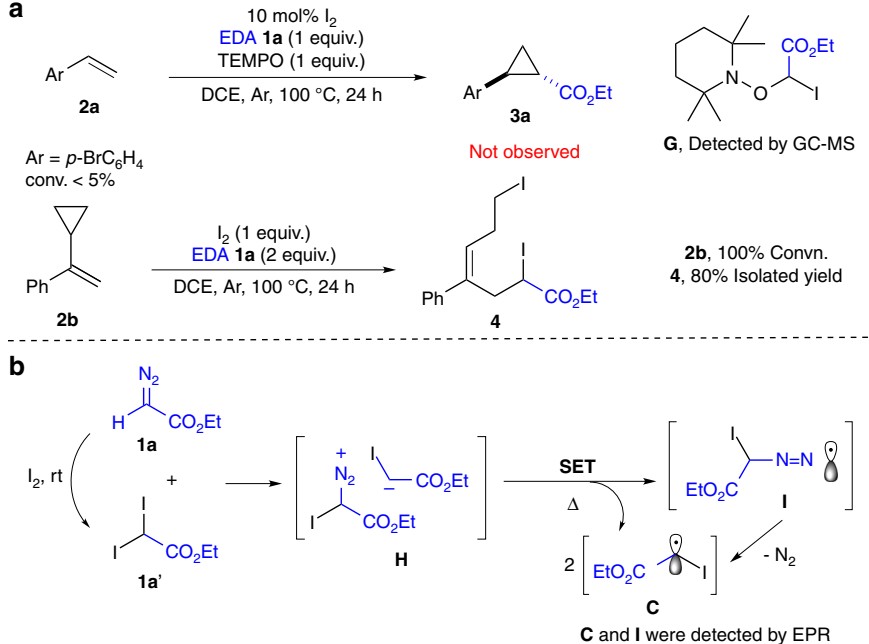

| Entry | Reactant | Additive | 2a conv. | 3a yield |
|---|---|---|---|---|
| 1 | **1a'** | none | 0% | ND |
| 2 | **1a** | none | <10% | <5% |
| 3 | **1a** | 10 mol% I₂ | 100% | 98% (96%) |
| 4 | **1a** | 1 equiv. of I₂ | 0% | ND |
| 5 | **1a** | 10 mol% **1a'** | 100% | 98% |
| 6 | **1a** | 10 mol% I₂, 80 °C, 48 h | <5% | <5% |
| 7 | **1a** | 10 mol% other catalysts (Br₂, KI, NIS) | <10% | <5% |

Reaction conditions: DCE solution (3 mL) of **2a** (0.3 mmol), additive and EDA **1a** (0.3 mmol) or **1a'** (0.3 mmol) was kept under argon (degassed) at 100 °C for 24 h. Conversion and yield were determined by $^1$H NMR spectroscopy using 1,3,5-trimethoxybenzene as an internal standard. Isolated yield is given within parentheses.
EDA ethyl diazoacetate, DCE 1,2-dichloroethane, $^1$H NMR proton nuclear magnetic resonance, ND not determined

**Fig. 4** Evidences for radical process under thermal condition. **a** Radical trapping and radical clock experiments. **b** Proposed mechanism for the generation of intermediate **C**

94% and 76% yields, respectively. Furthermore, vinyl ether (**3u**), methyl 2-phenylacrylate (**3v**), eneyne (**3w**), and diene (**3 s**) are all suitable for this new thermal condition, providing desired products in good yields. The conjugated diene and eneyne also gave excellent regio-selectivity. Other types of diazo compounds were also prepared and applied to this transformation. Interestingly, acceptor–acceptor-type diazo compound with a ketone substitution reacted with alkene and gave dihydrofuran **3×** through a sequential cyclopropane rearrangement. On the other hand, the donor–acceptor-type diazo compounds were inactive under this new condition likely due to the increased steric hindrance and competing carbene dimerization. Overall, this thermal approach for iodine-catalyzed diazo activation exhibits a very broad substrate scope. The fact that both electron-rich and electron-deficient alkenes works well under this condition is in sharp contrast with traditional metal carbenoid chemistry and more resembles the Co(II)–porphyrin-associated carbene radicals[8].

**Iodine-catalyzed pyrrole synthesis from enamides and diazo compounds**. To further demonstrate the uniqueness of this iodine-catalyzed diazo activation method, we focused on transformations that are not suitable with typical metal-catalyzed diazo substrates. As shown in Fig. 5, typical metal catalysts, such as Rh, Cu, Pd, and Au (see Supplementary Table 3), could not promote this cyclopropanation of enamide, giving low starting material conversion with EDA decomposition as the major products. To evaluate the new iodine-catalyzed diazo activation method, we

**Table 4 Reaction scope under thermal-initiated conditions**

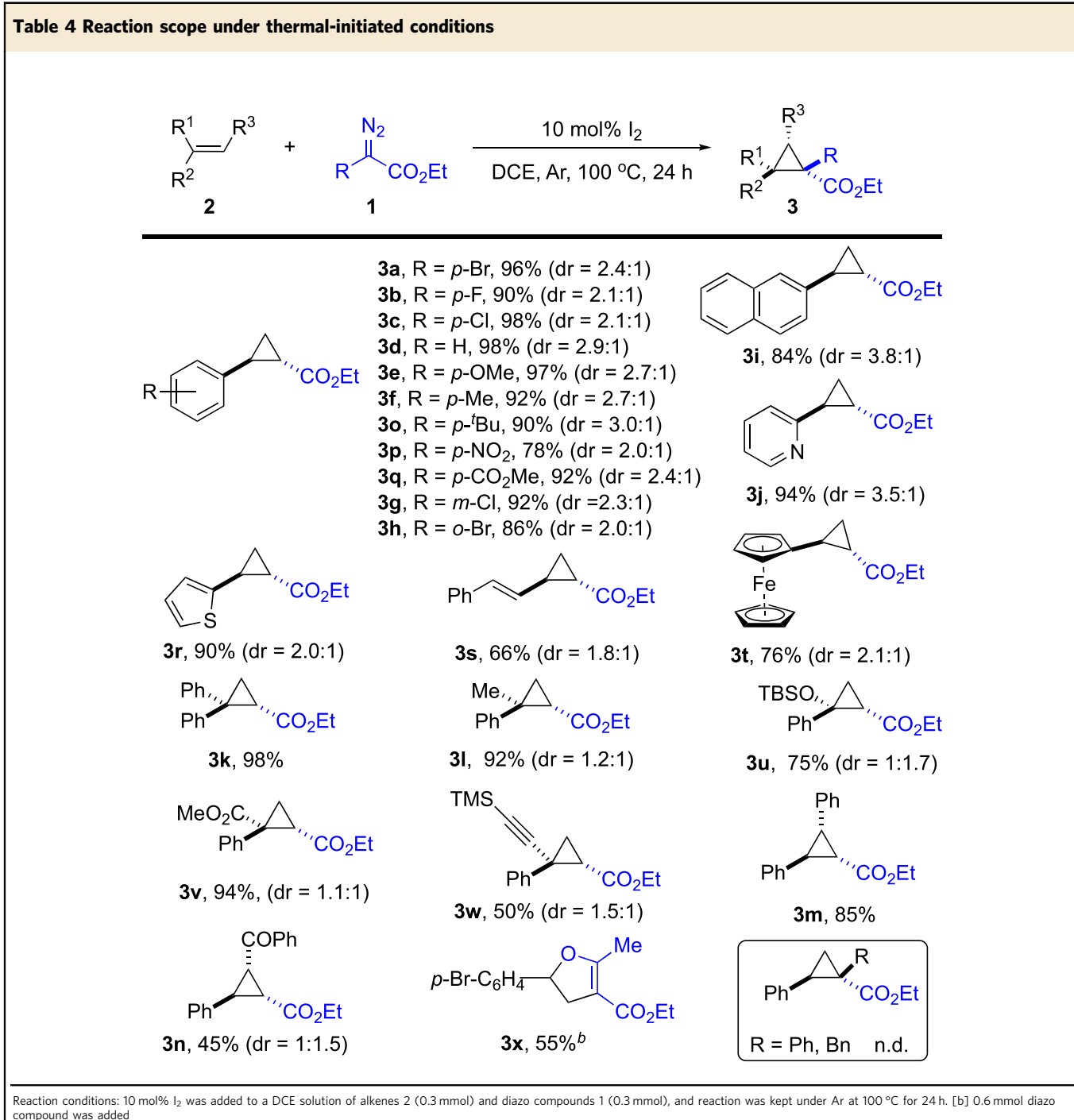

Reaction conditions: 10 mol% $I_2$ was added to a DCE solution of alkenes 2 (0.3 mmol) and diazo compounds 1 (0.3 mmol), and reaction was kept under Ar at 100 °C for 24 h. [b] 0.6 mmol diazo compound was added
*DCE* 1,2-dichloroethane, *ND* not determined, *dr* diastereomeric ratio

investigated this challenging transformation. To our delight, pyrrole **6a** was observed with 87% yield under thermal-initiation condition, while almost no reaction under photo-initiated conditions. Monitoring the reaction process revealed the formation of cyclopropane **6c′**, which would convert to **6c** through cyclopropane ring opening. Compound **6c′** was isolated and charged under the thermal conditions, giving the desired pyrrole **6c** in quantitative yield, which further confirmed the proposed reaction cascade. To the best of our knowledge, this is the first synthesis of pyrroles from enamides and diazo compounds.

The reaction scope of this new pyrrole synthesis is summarized in Table 5. Similar to the cyclopropanation, various aryl and alkyl enamides are well suited for this transformation, giving the corresponding pyrroles in good yields (**6a–6o**). Notably, the highly sterically congested tetrasubstituted pyrroles could also be formed in moderate yield, indicating the remarkable efficiency of this new transformation (**6p** and **6q**). Other diazo esters and ketones were applied to this reaction and gave the desired products in good to excellent yields (**6r–6x**). To showcase the practicality of this methodology, a gram-scale reaction was performed, and **6a** was obtained in 80% yield (1.83 g). Compared with previously reported metal-catalyzed formal [3 + 2] cycloaddition of enamide with alkyne in the presence of an oxidant[37–44], this iodine-catalyzed formal [4 + 1] cycloaddition of enamide and diazo compound is more practical under overall greener conditions (metal-free, oxidant-free, gram-scale synthesis, no

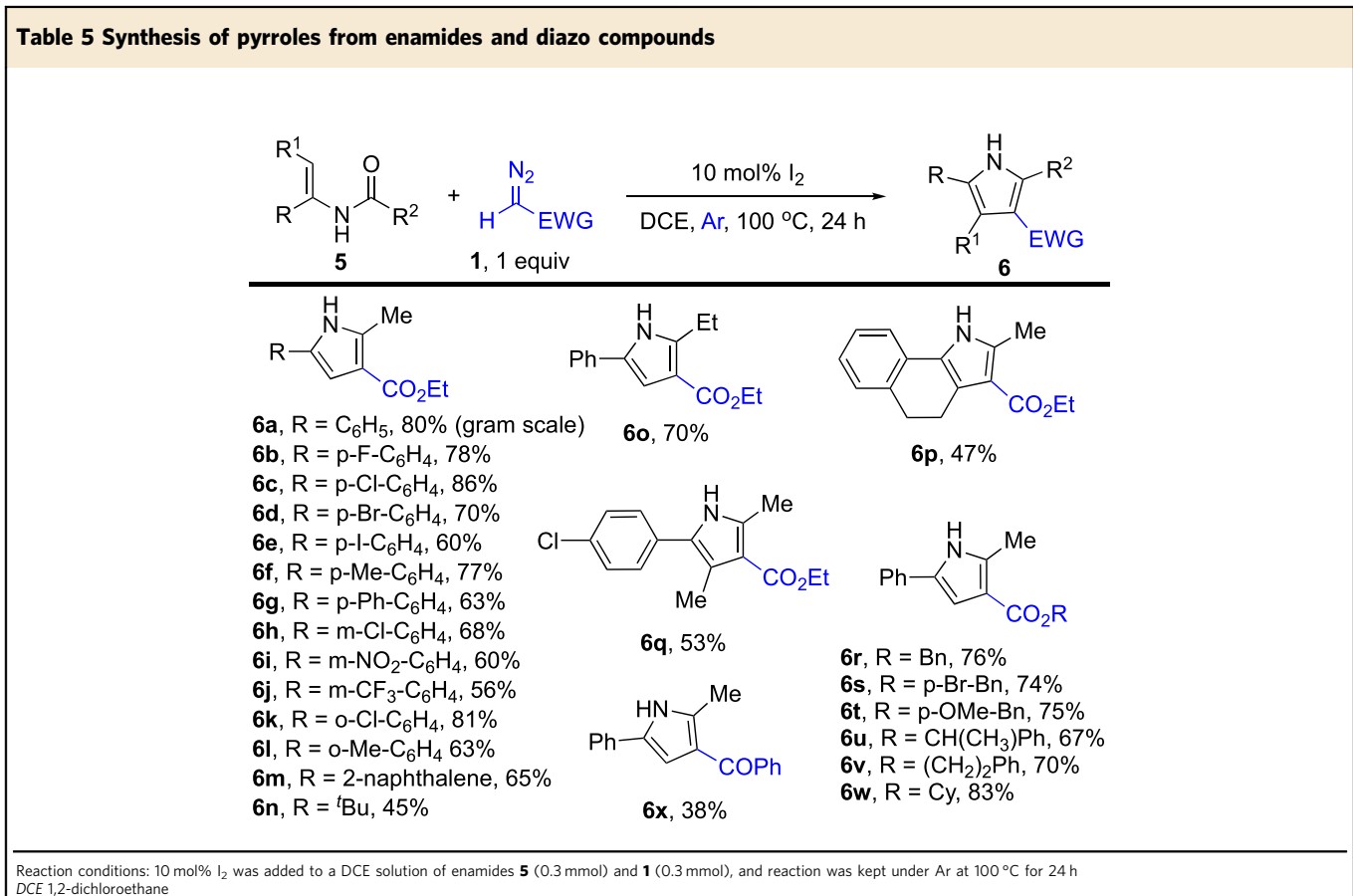

**Fig. 5** Thermally initiated iodine-catalyzed pyrrole synthesis. Generation pyrrole from the key cyclopropane intermediate **6'**

**Table 5 Synthesis of pyrroles from enamides and diazo compounds**

6a, R = C₆H₅, 80% (gram scale)
6b, R = p-F-C₆H₄, 78%
6c, R = p-Cl-C₆H₄, 86%
6d, R = p-Br-C₆H₄, 70%
6e, R = p-I-C₆H₄, 60%
6f, R = p-Me-C₆H₄, 77%
6g, R = p-Ph-C₆H₄, 63%
6h, R = m-Cl-C₆H₄, 68%
6i, R = m-NO₂-C₆H₄, 60%
6j, R = m-CF₃-C₆H₄, 56%
6k, R = o-Cl-C₆H₄, 81%
6l, R = o-Me-C₆H₄ 63%
6m, R = 2-naphthalene, 65%
6n, R = ᵗBu, 45%

6o, 70%

6p, 47%

6q, 53%

6r, R = Bn, 76%
6s, R = p-Br-Bn, 74%
6t, R = p-OMe-Bn, 75%
6u, R = CH(CH₃)Ph, 67%
6v, R = (CH₂)₂Ph, 70%
6w, R = Cy, 83%

6x, 38%

Reaction conditions: 10 mol% I₂ was added to a DCE solution of enamides **5** (0.3 mmol) and **1** (0.3 mmol), and reaction was kept under Ar at 100 °C for 24 h
*DCE* 1,2-dichloroethane

hazardous by-product). This successful construction of pyrroles from enamides greatly highlighted the advantage of this novel iodine-catalyzed radical approach in promoting challenging transformations.

**Iodine-catalyzed diazo activation toward olefin epoxidation.** Since the key iodo-substituted alkyl radical **C** was well defined by EPR and TEMPO experiment (Fig. 4), we wonder if the radical **C** could be trapped by O₂ to generate peroxide radical **J**. To our delight, peroxide radical was clearly observed via EPR experiment (see EPR spectra in Supplementary Fig. 5). Encouraged by this result, we hypothesized that peroxide radical **J** would react with olefin to form alkyl radical **K**, which could undergo fragmentation to generate the epoxide and ethyl 2-oxoacetate along with the regeneration of iodine catalyst (see Supplementary Fig. 6). With

this in mind, we investigated the reaction of olefin and EDA in the presence of 5 mol% I₂ catalyst in the open air (see details in Supplementary Table 3). To our delight, the desired epoxide **7a** was obtained in 70% yield, and the yield of **7a** was raised to 94% yield under O₂. No reaction occurred when treating alkene with EDA alone, indicating that I₂ catalyst was crucial for this reaction. Stoichiometric amount of EDA was needed for this reaction, although it was not part of the product **7a**. In addition, no desired product **7a** was formed at 60 °C, which suggested a plausible activation barrier for this reaction.

With this new epoxidation protocol, various alkenes were tested to investigate the scope of the reaction (see Supplementary Table 4). Olefin with electron-withdrawing groups (Br, Cl, NO₂, CO₂Me, and benzotriazole) worked well, giving the desired epoxide **7a–7e** in good to excellent yields. Olefin with electron-

withdrawing groups (OMe and Me) gave messy reactions under the optimal conditions (**3 f**), owing to the fast decomposition of the products in the presence of iodine.

## Discussion

In summary, we report herein a photo-initiated or thermal-initiated iodine-catalyzed diazo activation toward olefin cyclopropanation and epoxidation. An iodo-substituted alkyl radical was identified as the key intermediate experimentally. Significantly, the thermal-initiated and metal-free protocol provided broad substrate scope for cyclopropanation, which was highlighted by the highly-substituted pyrroles under thermal conditions. Further mechanistic study and synthetic applications of these intriguing iodo-substituted alkyl radicals are currently ongoing in our laboratory.

## Methods

**Olefin cyclopropanation under photo-initiated conditions**. To a 50 mL Schlenk tube with a stir bar was added olefin **2** (0.3 mmol), diazo compound **1** (0.3 mmol) and 3 mL of DCE, then Ru(bpy)$_3$Cl$_2$·6H$_2$O (2 mg, 0.003 mmol) and I$_2$ (8 mg, 0.03 mmol) were added. The Schlenk tube was vacuumed and purged with argon three times before it was tightly screw-capped. The reaction mixture was stirred at room temperature under blue light for 24 h. The reaction solution was evaporated, and the residue was purified by column chromatography (PE/EA) to afford the desired product **3a–3n**.

**Olefin cyclopropanation and pyrrole synthesis under thermal conditions**. To a 50 mL Schlenk tube with a stir bar was added olefin **2** (0.3 mmol) or enamides (0.3 mmol), diazo compound **1** (0.3 mmol) and 3 mL of DCE, then I$_2$ (8 mg, 0.03 mmol) were added. The Schlenk tube was vacuumed and purged with argon three times before it was tightly screw-capped. The reaction mixture was stirred at 100 °C for 24 h, and cooled to room temperature. The reaction solution was evaporated, and the residue was purified by column chromatography (PE/EA) to afford the desired product **3a–3x** and **5a–5t**.

**Olefin epoxidation under thermal-initiated conditions**. To a 50 mL Schlenk tube with a stir bar was added olefin **2** (0.3 mmol), diazo compound **1** (0.3 mmol) and 3 mL of DCE, then I$_2$ (4 mg, 0.015 mmol) were added. The Schlenk tube was vacuumed and purged with O$_2$ three times before it was tightly screw-capped. The reaction mixture was stirred at 80 °C for 2 h, and cooled to room temperature. The reaction solution was evaporated, and the residue was purified by column chromatography (PE/EA) to afford the desired product **7a–7e**.

**Data availability**. All data that support the findings of this study are available in the online version of this paper in the accompanying Supplementary Information (including experimental procedures, compound characterization data).

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

## Acknowledgements

We are grateful to the NSF (CHE-1665122), NIH (1R01GM120240-01), and NSFC (21228204, 21373246, and 21522309) for financial support.

## Author contributions

P.L., J.Z., L.S., and J.W. performed the experiments and prepared the Supplementary Information. X.S. and F.L. conceived and directed the project and wrote the paper.

## Additional information

**Competing interests:** The authors declare no competing interests.

