## [Peer Review File · Nature Communications]

Reviewer #1 (Remarks to the Author):

The manuscript by Shi, Li and coworkers reports on an iodine-catalysed de novo cyclopropanation of styrenes and related alkenes. The first section of the manuscript deals with the development of the cyclopropanation of styrenes under photoredox catalysis. To this end, the authors employ EDA as a stoichiometric (traceless) reagent, which represents a certain limitation regarding (commercial) availability of structural derivatives. However, the reaction is very clean and certainly superior to a recent protocol requiring a preformed geminal diiodinated reagent (ref. 22). The reported yields for styrene cyclopropanation from Table 2 are impressive. I agree with the authors that this is a very useful protocol. The mechanistic proposal from Figure 2 is balanced and corroborated by control experiments, particularly the radical clock experiment from Figure 3. I agree that the carbon-carbon bond formation from the benzylic radical at stage B' will be the route to cyclopropanation. Iodine-iodine interactions may already be present at this stage, however, the formation of the diiodine radical anion may also occur at a later stage as suggested by the authors. The initial photocatalytic protocol is then complimented by a cyclopropanation under thermal conditions (Tables 3, 4), and this variant provides an even better substrate scope. My major problem with this section is that the authors propose the same radical intermediary species C that is present under the photochemical conditions. If C is the immediate active species prior to alkene functionalisation, why is the substrate scope so different? Do the thermal reactions also proceed in the dark, or is daylight required for successful conversion?

These points should be addressed upon revision.

Finally, under thermal conditions, the authors include enamides as substrates to engage in dehydrative rearrangement of the corresponding 1-aminocyclopropanes thus generating pyrrole derivatives. This is a conceptually interesting pyrrole synthesis from readily available enamides and diazo esters. The presented scope is accurate.

The literature section is concise and well balanced. Overall, this is an extremely useful addition to the currently growing field of homogeneous iodine catalyses.

Minor points:

Throughout the Schemes, please use a uniform abbreviation for room temperature.

Reaction scheme from Table 2: maybe R1 or R2 should be labeled as Ar to illustrate that styrene derivatives are employed.

Page 4, line 60: the process to 1a' is not an iodine catalysis but a stoichiometric transformation.

Page 9, Figure 4: Please adjust the stoichiometry: from intermediate H, SET provides two intermediates C.

With respect to the SI, the characterization is largely fine. Regarding the ^{13}C NMR spectra, please provide full ranges to high fields of -5 ppm to fully demonstrate compound purity.

Reviewer #2 (Remarks to the Author):

This paper by Shi, Li & coworkers describes the I_2 catalyzed carbene transfer to olefins, both under thermal and photo-redox catalytic conditions, leading to cyclopropanation and pyrrole formation. Good yields are reported, and in case of the thermal reactions also a broad substrate scope is achieved. Products are well-characterized in a very detailed supporting information. The paper is very interesting and can be accepted after a few minor corrections:

- English can be improved in several places.
- It is not clear why the thermal reactions have a broader substrate scope than the reactions under photo-redox conditions, while almost the same mechanism is proposed. A decent explanation should be provided.
- Figure 5, conversion of cyclopropane 6' to pyrrole 6 would benefit from a proposed mechanism.
- Two recent publications from the de Bruin group about cobalt carbene radical chemistry could be added to refs 7-14, as they seem relevant for the work described in this paper: *Angew. Chem. Int. Ed.*, 2018, 57, 140-145 and *Chemical Science*, 2017, 8, 8221–8230.

Reviewer #3 (Remarks to the Author):

Shi, Li and co-workers developed a novel and attractive strategy for the olefin cyclopropanation through iodine catalyzed diazo activation under photo- or thermal-induced conditions. The reaction involved a plausible iodo-substituted alkyl radical as the key intermediate, which is interesting. However, the development of transition metal catalyzed carbene (come from diazo compounds) insertion into alkene to enable cyclopropanation is very mature, where high stereoselectivity could be achieved. In addition, the similar photo-initiated CH_2I_2 activated olefin cyclopropanation has been reported by Suero. (Figure 2B)

In general, I think the novelty of the manuscript is not enough and could not to be published in the Nature Communications.

Suggestions:

The ratio of E/Z in Table 2 and 4 should be defined as dr value.

Reviewer 1:

*Comments: The manuscript by Shi, Li and coworkers reports on an iodine-catalyzed de novo cyclopropanation of styrenes and related alkenes. The first section of the manuscript deals with the development of the cyclopropanation of styrenes under photoredox catalysis. To this end, the authors employ EDA as a stoichiometric (traceless) reagent, which represents a certain limitation regarding (commercial) availability of structural derivatives. However, the reaction is very clean and certainly superior to a recent protocol requiring a preformed geminal diiodinated reagent (ref. 22). The reported yields for styrene cyclopropanation from Table 2 are impressive. I agree with the authors that this is a very useful protocol. The mechanistic proposal from Figure 2 is balanced and corroborated by control experiments, particularly the radical clock experiment from Figure 3. I agree that the carbon-carbon bond formation from the benzylic radical at stage **B'** will be the route to cyclopropanation.*

Iodine-iodine interactions may already be present at this stage, however, the formation of the diiodine radical anion may also occur at a later stage as suggested by the authors. The initial photocatalytic protocol is then complimented by a cyclopropanation under thermal conditions (Tables 3, 4), and this variant provides an even better substrate scope.

*Question 1: My major problem with this section is that the authors propose the same radical intermediary species **C** that is present under the photochemical conditions. If **C** is the immediate active species prior to alkene functionalization, why is the substrate scope so different? Do the thermal reactions also proceed in the dark, or is daylight required for successful conversion? These points should be addressed upon revision.*

Response: We thank the reviewer for these very positive comments and insightful questions. This is a very good question! In fact, the same radical intermediate **C** was formed in both thermal and photo-chemical conditions based on the TEMPO experiments and radical clock experiments as shown in our previous submitted SI. The key reason for different substrate performance under photo or thermal conditions was the fact that certain alkenes (vinyl ferrocene, *p*-NO₂-styrene or enamide) could quench the photocatalytic reaction. As shown in the figure below, under the photocatalytic conditions, *p*-Br-styrene gave cyclopropane product in 96% NMR yields. However, upon addition of vinyl ferrocene, *p*-NO₂-styrene or enamide, this reaction was quenched with no desired product obtained. (These experiments have been added in the revised SI for clarification)

Further exploration revealed that the key diiodide intermediate **1a'** undergoes slow decomposition caused by photocatalyst. As a result, the starting material EDA slowly decomposed due to the formation of I₂ upon **1a'** decomposition. Therefore, no **1a'** have been obtained under the photocatalytic condition. In contrast, addition of these alkenes (vinyl ferrocene, *p*-NO₂-styrene and enamide) prevented photocatalyst promoted **1a'** decomposition. Thus, addition of these alkenes into the reaction mixture, **1a'** was obtained as shown below.

With above reasons, alkenes, such as vinyl ferrocene, *p*-NO₂-styrene and enamide, only work under the thermal conditions.

Two other alkenes (diene and vinyl ether) are not compatible under photo conditions. This is because of the alkenes decomposition in the presence of photocatalyst. As shown in the figure below, significant alkene decomposition was observed owing to vigorous alkene polymerization.

To confirm that thermal conditions does not require light, we carried out the thermal experiment under dark conditions. The same result was obtained as under daylight conditions, which proved that light is not required for the generation of radical intermediate **C** under thermal conditions.

Overall, this is a very good question from the reviewer and we greatly appreciate the insightful questions. All these control experiments are added in the revised SI and a discussion was added in the revised manuscript.

Finally, under thermal conditions, the authors include enamides as substrates to engage in dehydrative rearrangement of the corresponding 1-aminocyclopropanes thus generating pyrrole derivatives. This is a conceptually interesting pyrrole synthesis from readily available enamides and diazo esters. The presented scope is accurate.

We are grateful for the reviewer's recognition of the significance of this work. This also partially answered reviewer #3's concern on the strong impact of this **metal-free condition** over traditional metal catalyzed conditions

The literature section is concise and well balanced. Overall, this is an extremely useful addition to the currently growing field of homogeneous iodine catalyzt.

Minor points:

Throughout the Schemes, please use a uniform abbreviation for room temperature.

Response: Thanks reviewer very much for this reminder. All inconsistent “room temperature” in the original Schemes have been abbreviated as “r.t.”. Please check the revised manuscript for details.

Reaction scheme from Table 2: maybe R1 or R2 should be labeled as Ar to illustrate that styrene derivatives are employed.

Response: We thank the reviewer for careful evaluation of our substrate scope. According to the reviewer’s suggestion, R² in the Table 2 has been labeled as Ar, and R³ have been changed to R² to better represents the substrate scope of this table. Please check the revised manuscript for details.

Page 4, line 60: the process to **1a'** is not an iodine catalysis but a stoichiometric transformation.

Response: Thanks reviewer for this very important reminder. Yes, the generation of diiodo compound **1a'** need stoichiometric iodine, therefore, original “With the realization of diiodo compound **1a'** formation through iodine catalyzed diazo activation under mild conditions” has been changed to “With the realization of diiodo compound **1a'** by reacting diazo compound with I₂ under mild conditions” for clarification as suggested.

Page 9, Figure 4: Please adjust the stoichiometry: from intermediate H, SET provides two intermediates C.

Response: Thanks reviewer for this important comment. Figure 4 has been revised as shown below.

With respect to the SI, the characterization is largely fine. Regarding the ^{13}C NMR spectra, please provide full ranges to high fields of -5 ppm to fully demonstrate compound purity.

Response: According to reviewer's suggestion, we have provided full ranges (-5 ppm to 200 ppm) of ^{13}C NMR spectra, which clearly show the high purity of all products.

Reviewer 2:

This paper by Shi, Li & coworkers describes the I_2 catalyzed carbene transfer to olefins, both under thermal and photo-redox catalytic conditions, leading to cyclopropanation and pyrrole formation. Good yields are reported, and in case of the thermal reactions also a broad substrate scope is achieved. Products are well-characterized in a very detailed supporting information. **The paper is very interesting and can be accepted after a few minor corrections.**

English can be improved in several places.

Response: We have refined our English composition throughout the manuscript and we apologize for the inconvenience that caused by our improficiency in language. Efforts have been made through proof reading by English native speaker. All changes have been highlighted in the manuscript.

It is not clear why the thermal reactions have a broader substrate scope than the reactions under photo-redox conditions, while almost the same mechanism is proposed. A decent explanation should be provided.

Response: This is the same concern as reviewer #1 pointed. We are grateful both reviewers caught this important point. A detailed explanation has been given as shown above. The detailed changes are highlighted in the revised manuscript and revised supporting information.

Figure 5, conversion of cyclopropane **6'** to pyrrole **6** would benefit from a proposed mechanism.

Response: A very good point! A detailed arrow pushing mechanism is shown in the revised mechanism in Figure 5.

Two recent publications from the de Bruin group about cobalt carbene radical chemistry could be added to refs 7-14, as they seem relevant for the work described in this paper: Angew. Chem. Int. Ed., 2018, 57, 140-145 and Chemical Science, 2017, 8, 8221–8230.

Response: The work from Bruin group represents very important advances in metal carbene radical chemistry; we appreciate the reviewer's help for pointing it out. These two publications have been added as ref. 15 and 16 as suggested.

Reviewer 3:

Shi, Li and co-workers developed a novel and attractive strategy for the olefin cyclopropanation through iodine catalyzed diazo activation under photo- or thermal-induced conditions. The reaction involved a plausible iodo-substituted alkyl radical as the key intermediate, which is interesting. However, the development of transition metal catalyzed carbene (come from diazo compounds) insertion into alkene to enable cyclopropanation is very mature, where high stereoselectivity could be achieved. In addition, the similar photo-initiated CH₂I₂ activated olefin cyclopropanation has been reported by Suero. (Figure 2B)

In general, I think the novelty of the manuscript is not enough and could not to be published in the Nature Communications.

Response: We thank the reviewer for recognizing the innovation of this iodine initiated diazo activation to access radicals. However, we respectfully disagree the novelty concern raised by this reviewer.

First, this work is about the iodine catalyzed diazo activation through a radical pathway. Not only the reaction condition was new, but also the mechanistic was unprecedented in literature. As researchers are all generally believed that the advancement of chemical synthesis basically relies on the understanding of reaction mechanism and any fundamentally different reaction path will almost surely initiate exploration of new direction. With the high regards of the journal of Nature Communication, this is exactly why we are so excited about this work and strongly believed (also confirmed by the other two reviewers) that the novelty of this work fits in a publication like Nat. Commun.

Second, reviewer concern that metal catalyzed cyclopropanation conditions have been well studied in the past. However, this is exactly the other important point of this work. With the increasing concerns on environmental safety, performing reactions under mild and environmental friendly (greener) conditions is a general goal for modern chemical synthesis. Thus, an effective non-metal catalyzed protocol that substituted previous metal catalysis is generally considered as an important advancement. While we fully aware the application of Ru photo catalysts for this transformation, the thermal condition is certainly more practical and makes the transformation greener. With all respect to the reviewer, we consider the discovery of a metal-free variant for a traditional metal catalyzed transformation is a great improvement.

Finally, the concern of similarity with Suero's report was not accurate. Although similar iodo-substituted alkyl radicals were generated in both our and Suero's cases, there are significant differences between these two systems. In fact, the basic redox mechanisms are total different between the two systems. First, in Suero's case, a preformed diiodide is required to initiate this reaction, while in our case diiodide can be generated *in situ* by treating diazo compound with catalytic I_2 , which dramatically expanded the substrate scope due to the readily availability of the diazo compounds. Second, in Suero's case, a reductive quenching cycle (Ru^{II}/Ru^I cycle) occurred, so large excess of reductant is needed to generate the iodo-substituted alkyl radical as well as quench the iodine generated after cyclopropane formation (5 eq $i\text{-Pr}_2\text{EtN}$ and $\text{Na}_2\text{S}_2\text{O}_3$, respectively); while in our case, an oxidative quenching cycle (Ru^{II}/Ru^{III} cycle) took place, and only catalytic amount of I_2 is needed to complete the catalytic cycle. In fact, these two crucial features of our system are the foundation for the success catalytic activation that could not been

obtained in Suero's work and made the reported reaction herein a more atom-economical, practical and efficient new system.

Notably, as reviewer #1 pointed out, the synthesis of substituted pyrrole does not work under any metal catalyzed conditions and *"This is a conceptually interesting pyrrole synthesis from readily available enamides and diazo esters"*.

With the great respect to the journal and all the reviewers, we further extended this chemistry to epoxidation, another novel diazo transformation that have NEVER been reported and could not be promoted by transition metal catalyst.

We have discovered that this iodo-substituted alkyl radical can be successfully trapped by molecular oxygen; the peroxide radical **J** can react with alkene to yield benzyl radical **K**, which upon fragmentation to give epoxide product. Although the substrate scope of this epoxidation is limited to electron-deficient alkenes, it clearly showed another unique reactivity of the iodo-substituted alkyl radical. Details of this part of work is included in the revised manuscript.

Our proposal: iodine catalyzed diazo activation toward olefin epoxidation

Suggestions: The ratio of E/Z in Table 2 and 4 should be defined as dr value.

Response: We thank the reviewer's helpful suggestion. According to the reviewer's suggestion, the ratio of E/Z in Table 2 and 4 of the manuscript has been defined as dr value.

Overall, we truly hope with the delicate revision of our manuscript the reviewer will agree on the novelty of this work, and provide positive respond for publishing this manuscript on *Nature Commnucation*.

Reviewer #1 (Remarks to the Author):

The revised manuscript by Shi, Li and coworkers has fully met all my earlier requests for revision and have explained my concerns regarding the mechanism and the behavior of intermediate C. I am convinced that the work will have a high impact in the emerging field of iodine catalysis.

During the revision, the authors have included a section on aerobic epoxidation, in which the diazo ester is a sacrificial reagent. This reaction arises from interception of the iodinated acetic ester by dioxygen. This is quite a different mode of reaction from 1a' and I am not convinced that it fits very well into the present communication. I found the earlier communication focusing on cyclopropanation alone a more concise message. However, I agree that the results are worthy of interest and I leave it to the authors and the editors to decide whether they should remain or be removed.

Reviewer #2 (Remarks to the Author):

In my view the authors addressed all important issues raised in the first review round. I feel this made the paper stronger and in my view the paper can be accepted now.